# OmniH2O: Universal and Dexterous Human-to-Humanoid Whole-Body Teleoperation and Learning

**Tairan He**[†1]**, Zhengyi Luo**[†1]**, Xialin He**[†2]**, Wenli Xiao**[1]**, Chong Zhang**[1]**,
**Weinan Zhang**[2]**, Kris Kitani**[1]**, Changliu Liu**[1]**, Guanya Shi**[1]

[1]Carnegie Mellon University    [2]Shanghai Jiao Tong University    [†]Equal Contributions
Page: https://omni.human2humanoid.com
Code: https://github.com/LeCAR-Lab/human2humanoid

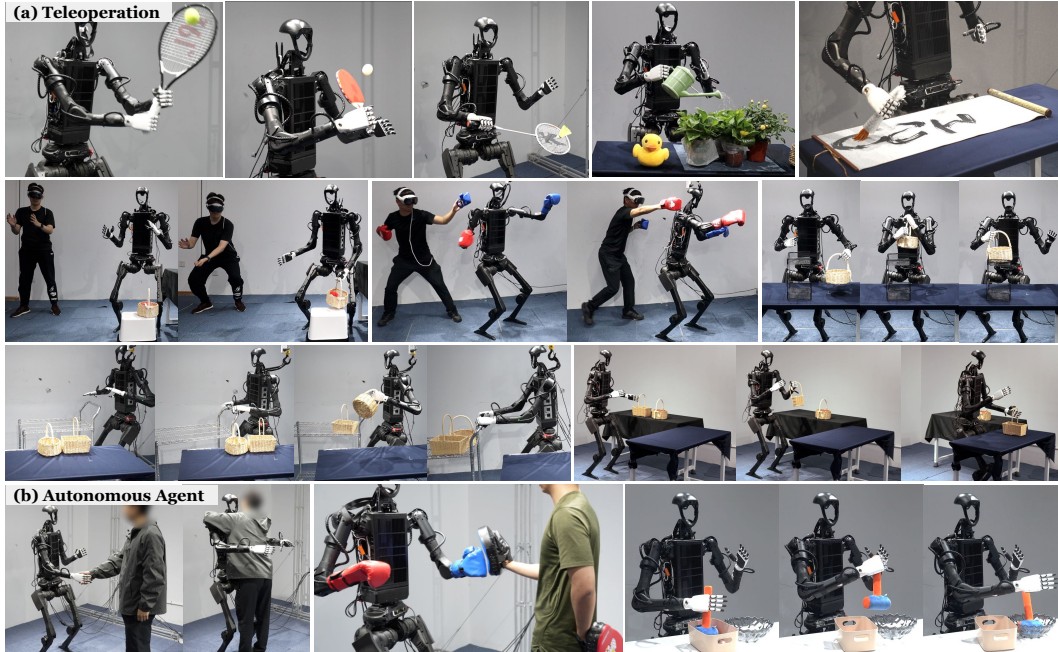

Figure 1:  (a) OmniH2O enables teleoperating a full-size humanoid robot (Unitree H1) to complete tasks that require both high-precision manipulation and locomotion.  (b) OmniH2O also enables full autonomy through visual input, controlled by GPT-4o or a policy learned from teleoperated demonstrations. **Videos**: see our website: https://omni.human2humanoid.com

**Abstract:** We present OmniH2O (Omni Human-to-Humanoid), a learning-based system for whole-body humanoid teleoperation and autonomy. Using kinematic pose as a universal control interface, OmniH2O enables various ways for a human to control a full-sized humanoid with dexterous hands, including using real-time teleoperation through VR headset, verbal instruction, and RGB camera. OmniH2O also enables full autonomy by learning from teleoperated demonstrations or integrating with frontier models such as GPT-4o. OmniH2O demonstrates versatility and dexterity in various real-world whole-body tasks through teleoperation or autonomy, such as playing multiple sports, moving and manipulating objects, and interacting with humans, as shown in  Figure 1. We develop an RL-based sim-to-real pipeline, which involves large-scale retargeting and augmentation of human motion datasets, learning a real-world deployable policy with sparse sensor input by imitating a privileged teacher policy, and reward designs to enhance robustness and stability. We release the first humanoid whole-body control dataset, *OmniH2O-6*, containing six everyday tasks, and demonstrate humanoid whole-body skill learning from teleoperated datasets.

**Keywords:** Humanoid Teleoperation, Humanoid Loco-Manipulation, RL

8th Conference on Robot Learning (CoRL 2024), Munich, Germany.

# 1 Introduction

How can we best unlock humanoid's potential as one of the most promising physical embodiments of general intelligence? Inspired by the recent success of pretrained vision and language models [1], one potential answer is to collect large-scale human demonstration data in the real world and learn humanoid skills from it. The embodiment alignment between humanoids and humans not only makes the humanoid a potential generalist platform but also enables the seamless integration of human cognitive skills for scalable data collections [2, 3, 4, 5].

However, *whole-body control* of a full-sized humanoid robot is challenging [6], with many existing works focusing only on the lower body [7, 8, 9, 10, 11, 12, 13] or decoupled lower and upper body control [14, 4, 15]. To simultaneously support stable dexterous manipulation and robust locomotion, the controller must coordinate the lower and upper bodies in unison. For the humanoid *teleoperation interface* [2], the need for expensive setups such as motion captures and exoskeletons also hinders large-scale humanoid data collection. In short, we need a robust control policy that *supports whole-body dexterous loco-manipulation*, while seamlessly integrating with *easy-to-use and accessible teleoperation interfaces* (e.g., VR) to enable scalable demonstration data collection.

In this work, we propose OmniH2O, a learning-based system for whole-body humanoid teleoperation and autonomy. We propose a pipeline to train a robust whole-body motion imitation policy via teach-student distillation and identify key factors in obtaining a stable control policy that supports dexterous manipulation. For instance, we find these elements to be essential: *motion data distribution*, *reward designs*, and *state space design* and *history utilization*. The distribution of the motion imitation dataset needs to be biased toward standing and squatting to help the policy learn to stabilize the lower body during manipulation. Regularization rewards are used to shape the desired motion but need to be applied with a curriculum. The input history could replace the global linear velocity, an essential input in previous work [3] that requires Motion Capture (MoCap) to obtain. We also carefully design our control interface and choose the kinematic pose as an intermediate representation to bridge between human instructions and humanoid actuation. This interface makes our control framework compatible with many real-world input sources, such as VR, RGB cameras, and autonomous agents (GPT-4o). Powered by our robust control policy, we demonstrate teleoperating humanoids to perform various daily tasks (racket swinging, flower watering, brush writing, squatting and picking, boxing, basket delivery, *etc*.), as shown in Figure 1. Through teleoperation, we collect a dataset of our humanoid completing six tasks such as hammer catching, basket picking, *etc*., annotated with paired first-person RGBD camera views, control input, and whole-body motor actions. Based on the dataset, we showcase training autonomous policies via imitation learning.

In conclusion, our contributions are as follows: **(1)** We propose a pipeline to train a robust humanoid control policy that supports whole-body dexterous loco-manipulation with a universal interface that enables versatile human control and autonomy. **(2)** Experiments of large-scale motion tracking in simulation and the real world validate the superior motion imitation capability of OmniH2O. **(3)** We contribute the first humanoid loco-manipulation dataset and evaluate imitation learning methods on it to demonstrate humanoid whole-body skill learning from teleoperated datasets.

# 2 Related Works

**Learning-based Humanoid Control**. Controlling humanoid robots is a long-standing robotic problem due to their high degree-of-freedom (DoF) and lack of self-stabilization [16, 17]. Recently, learning-based methods have shown promising results [7, 8, 9, 10, 11, 12, 13, 3, 18]. However, most studies [7, 8, 9, 10, 11, 12, 13] focus mainly on learning robust locomotion policies and do not fully unlock all the abilities of humanoids. For tasks that require whole-body loco-manipulation, the lower body must serve as the support for versatile and precise upper body movement [19]. Traditional goal-reaching [20, 21] or velocity-tracking objectives [18] used in legged locomotion are incompatible with such requirements because these objectives require additional task-specific lower-body goals (from other policies) to indirectly account for upper-lower-body coordination. OmniH2O instead learns an end-to-end whole-body policy to coordinate upper and lower bodies.

**Humanoid Teleoperation**. Teleoperating humanoids holds great potential in unlocking the full capabilities of the humanoid system. Prior efforts in humanoid teleoperation have used task-space control [4, 22], upper-body-retargeted teleoperation [23, 24] and whole-body teleoperation [25, 26, 27, 28, 29, 30, 3, 4]. Recently, H2O [3] presents an RL-based whole-body teleoperation framework that uses a third-person RGB camera to obtain full-body keypoints of the human teleoperator. However, due to the delay and inaccuracy of RGB-based pose estimation and the requirement for global linear velocity estimation, H2O [3] requires MoCap during test time, only supports simple mobility tasks, and lacks the precision for dexterous manipulation tasks. By contrast, OmniH2O enables high-precision dexterous loco-manipulation indoors and in the wild.

**Whole-body Humanoid Control Interfaces**. To control a full-sized humanoid, many interfaces such as exoskeleton [31], MoCap [32, 33], and VR [34, 35] are proposed. Recently, VR-based humanoid control [36, 37, 38, 39] has been drawing attention in the graphics community due to its ability to create whole-body motion using sparse input. However, these VR-based works only focus on humanoid control for animation and do not support mobile manipulation. OmniH2O, on the other hand, can control a real humanoid robot to complete real-world manipulation tasks.

**Open-sourced Robotic Dataset and Imitation Learning**. One major challenge within the robotics community is the limited number of publicly available datasets compared to those for language and vision tasks [40]. Recent efforts [40, 41, 42, 43, 44, 45, 46, 47] have focused on collecting robotic data using various embodiments for different tasks. However, most of these datasets are collected with fixed-base robotic arm platforms. Even one of the most comprehensive datasets to date, Open X-Embodiment [40], does not include data for humanoids. To the best of our knowledge, we are the first to release a dataset for full-sized humanoid whole-body loco-manipulation.

## 3 Universal and Dexterous Human-to-Humanoid Whole-Body Control

In this section, we describe our whole-body control system to support teleoperation, dexterous manipulation, and data collection. As simulation has access to inputs that are hard to obtain from real-world devices, we opt to use a teacher-student framework. We also provide details about key elements to obtain a stable and robust control policy: dataset balance, reward designs, *etc*.

**Problem Formulation**. We formulate the learning problem as goal-conditioned RL for a Markov Decision Process (MDP) defined by the tuple $\mathcal{M} = \langle \mathcal{S}, \mathcal{A}, \mathcal{T}, \mathcal{R}, \gamma \rangle$ of state $\mathcal{S}$, action $\boldsymbol{a}_t \in \mathcal{A}$, transition $\mathcal{T}$, reward function $\mathcal{R}$, and discount factor $\gamma$. The state $\boldsymbol{s}_t$ contains the proprioception $\boldsymbol{s}_t^{\text{p}}$ and the goal state $\boldsymbol{s}_t^{\text{g}}$. The goal state $\boldsymbol{s}_t^{\text{g}}$ includes the motion goals from the human teleoperator or autonomous agents. Based on proprioception $\boldsymbol{s}_t^{\text{p}}$, goal state $\boldsymbol{s}_t^{\text{g}}$, and action $\boldsymbol{a}_t$, we define the reward $r_t = \mathcal{R}\left(\boldsymbol{s}_t^{\text{p}}, \boldsymbol{s}_t^{\text{g}}, \boldsymbol{a}_t\right)$. The action $\boldsymbol{a}_t$ specifies the target joint angles and a PD controller actuates the motors. We apply the Proximal Policy Optimization algorithm (PPO) [48] to maximize the cumulative discounted reward $\mathbb{E}\left[\sum_{t=1}^{T} \gamma^{t-1} r_t\right]$. In this work, we study the motion imitation task where our policy $\pi_{\text{OmniH2O}}$ is trained to track real-time motion input as shown in Figure 3. This task provides a universal interface for humanoid control as the kinematic pose can be provided by many different sources. We define kinematic pose as $\boldsymbol{q}_t \triangleq (\boldsymbol{\theta}_t, \boldsymbol{p}_t)$, consisting of 3D joint rotations $\boldsymbol{\theta}_t$ and positions $\boldsymbol{p}_t$ of all joints on the humanoid. To define velocities $\dot{\boldsymbol{q}}_{1:T}$, we have $\dot{\boldsymbol{q}}_t \triangleq (\boldsymbol{\omega}_t, \boldsymbol{v}_t)$ as angular $\boldsymbol{\omega}_t$ and linear velocities $\boldsymbol{v}_t$. As a notation convention, we use $\tilde{\cdot}$ to represent kinematic quantities from VR headset or pose generators, $\hat{\cdot}$ to denote ground truth quantities from MoCap datasets, and normal symbols without accents for values from the physics simulation or real robot.

**Human Motion Retargeting**. We train our motion imitation policy using retargeted motions from the AMASS [49] dataset, using a similar retargeting process as H2O [3]. One major drawback of H2O is that the humanoid tends to take small adjustment steps instead of standing still. In order to enhance the ability of stable standing and squatting, we bias our training data by adding sequences that contain fixed lower body motion. Specifically, for each motion sequence $\hat{\boldsymbol{q}}_{1:T}$ from our

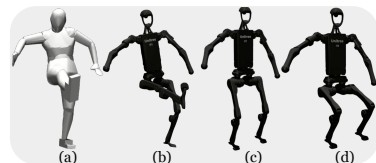

Figure 2: (a) Source motion; (b) Retargeted motion; (c) Standing variant; (d) Squatting variant.

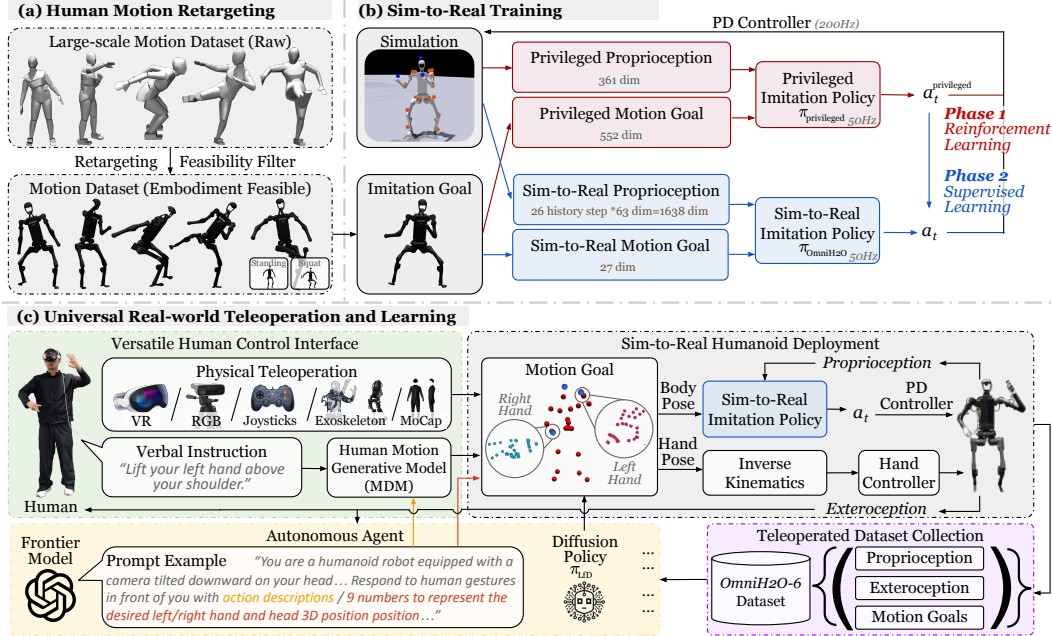

Figure 3: (a) OmniH2O retargets large-scale human motions and filters out infeasible motions for humanoids. (b) Our **sim-to-real policy** is distilled through supervised learning from an **RL-trained teacher policy** using privileged information. (c) The universal design of OmniH2O supports versatile **human control interfaces** including VR headset, RGB camera, language, *etc.* Our system also supports to be controlled by **autonomous agents** like GPT-4o or imitation learning policy trained using our **dataset collected via teleoperation**.

dataset, we create a "stable" version $\hat{q}_{1:T}^{\text{stable}}$ by fixing the root position and the lower body to a standing or squatting position as shown in Fig. 2. We provide ablation of this strategy in Appendix H.

**Reward and Domain Randomization**. To train $\pi_{\text{privileged}}$ that is suitable as a teacher for a real-world deployable student policy, we employ both imitation rewards and regularization rewards. Previous work [18, 3] often uses regularization rewards like *feet air time* or *feet height* to shape the lower-body motions. However, these rewards result in the humanoid stomping to keep balanced instead of standing still. To encourage standing still and taking large steps during locomotion, we propose a key reward function *max feet height for each step*. The *max feet height for each step* reward is designed to encourage the robot to take higher steps by rewarding it based on the maximum height achieved by the feet during the air phase of each step. We find that this reward, when applied with a carefully designed curriculum, effectively helps RL decide when to stand or walk. We provide a detailed overview of rewards, curriculum design, and domain randomization in Appendices E and F.

**Teacher: Privileged Imitation Policy**. During real-world teleoperation of a humanoid robot, much information that is accessible in simulation (*e.g.*, the global linear/angular velocity of every body link) is not available. Moreover, the input to a teleoperation system could be *sparse* (*e.g.*, for VR-based teleoperation, only the hands and head's poses are known), which makes the RL optimization challenging. To tackle this issue, We first train a teacher policy that uses privileged state information and then distill it to a student policy with limited state space. Having access to the privileged state can help RL find more optimal solutions, as shown in prior works [50] and our experiments ( Section 4). Formally, we train a privileged motion imitator $\pi_{\text{privileged}}(\boldsymbol{a}_t|\boldsymbol{s}_t^{\text{p-privileged}}, \boldsymbol{s}_t^{\text{g-privileged}})$, as described in Figure 3. The proprioception is defined as $\boldsymbol{s}_t^{\text{p-privileged}} \triangleq [\boldsymbol{p}_t, \boldsymbol{\theta}_t, \dot{\boldsymbol{q}}_t, \boldsymbol{\omega}_t, \boldsymbol{a}_{t-1}]$, which contains the humanoid rigidbody position $\boldsymbol{p}_t$, orientation $\boldsymbol{\theta}_t$, linear velocity $\dot{\boldsymbol{q}}_t$, angular velocity $\boldsymbol{\omega}_t$, and the previous action $\boldsymbol{a}_{t-1}$. The goal state is defined as $\boldsymbol{s}_t^{\text{g-privileged}} \triangleq [\hat{\boldsymbol{\theta}}_{t+1} \ominus \boldsymbol{\theta}_t, \hat{\boldsymbol{p}}_{t+1} - \boldsymbol{p}_t, \hat{\boldsymbol{v}}_{t+1} - \boldsymbol{v}_t, \hat{\boldsymbol{\omega}}_{t+1} - \boldsymbol{\omega}_t, \hat{\boldsymbol{\theta}}_{t+1}, \hat{\boldsymbol{p}}_{t+1}]$, which contains the reference pose $(\hat{\boldsymbol{\theta}}_t, \hat{\boldsymbol{p}}_t)$ and one-frame difference between the reference and current state for all rigid bodies of the humanoid.

**Student: Sim-to-Real Imitation Policy with History**. We design our control policy to be compatible with many input sources by using the kinematic reference motion as the intermediate representation. As estimating full-body motion $\tilde{\boldsymbol{q}}_t$ (both rotation and translation) is difficult (especially from

VR headsets), we opt to control our humanoid with position $\tilde{p}_t$ only for teleoperation. Specifically, for real-world teleoperation, the goal state is $s_t^{\text{g-real}} \triangleq (\tilde{p}_t^{\text{real}} - p_t^{\text{real}}, \tilde{v}_t^{\text{real}} - v_t^{\text{real}}, \tilde{p}_t^{\text{real}})$. The superscript $^{\text{real}}$ indicates using the 3-points available (head and hands) from the VR headset. For other control interfaces (e.g., RGB, language), we use the same input 3-point input to maintain consistency, though can be easily extended to more keypoints to alleviate ambiguity. For proprioception, the student policy $s_t^{\text{p-real}} \triangleq (d_{t-25:t}, \dot{d}_{t-25:t}, \omega_{t-25:t}^{\text{root}}, g_{t-25:t}, a_{t-25-1:t-1})$ uses values easily accessible in the real-world, which includes 25-step history of joint (DoF) position $d_{t-25:t}$, joint velocity $\dot{d}_{t-25:t}$, root angular velocity $\omega_{t-25:t}^{\text{root}}$, root gravity $g_{t-25:t}$, and previous actions $a_{t-25-1:t-1}$. The inclusion of history data helps improve the robustness of the policy with our teacher-student supervised learning. Note that no global linear velocity $v_t$ information is included in our observations and the policy implicitly learns velocity using history information. This removes the need for MoCap as in H2O [3] and further enhances the feasibility of in-the-wild deployment.

**Policy Distillation**. We train our deployable teleoperation policy $\pi_{\text{OmniH2O}}$ following the DAgger [51] framework: for each episode, we roll out the student policy $\pi_{\text{OmniH2O}}(a_t | s_t^{\text{p-real}}, s_t^{\text{g-real}})$ in simulation to obtain trajectories of $(s_{1:T}^{\text{p-real}}, s_{1:T}^{\text{g-real}})$. Using the reference pose $\hat{q}_{1:T}$ and simulated humanoid states $s_{1:T}^{\text{p}}$, we can compute the privileged states $s_t^{\text{g-privileged}}, s_t^{\text{p-privileged}} \leftarrow (s_t^{\text{p}}, \hat{q}_{t+1})$. Then, using the pair $(s_t^{\text{p-privileged}}, s_t^{\text{g-privileged}})$, we query the teacher $\pi_{\text{privileged}}(a_t^{\text{privileged}} | s_t^{\text{p-privileged}}, s_t^{\text{g-privileged}})$ to calculate the reference action $a_t^{\text{privileged}}$. To update $\pi_{\text{OmniH2O}}$, the loss is: $\mathcal{L} = \|a_t^{\text{privileged}} - a_t\|_2^2$.

**Dexterous Hands Control**. As shown in Figure 3(c), we use the hand poses estimated by VR [52, 53], and directly compute joint targets based on inverse kinematics for an off-the-shelf low-level hand controller. We use VR for the dexterous hand control in this work, but the hand pose estimation could be replaced by other interfaces (e.g., MoCap gloves [54] or RGB cameras [55]) as well.

## 4 Experimental Results

In our experiments, we aim to answer the following questions. **Q1.** (Section 4.1) Can OmniH2O accurately track motion in simulation and real world? **Q2.** (Section 4.2) Does OmniH2O support versatile control interfaces in the real world and unlock new capabilities of loco-manipulation? **Q3.** (Section 4.3) Can we use OmniH2O to collect data and learn autonomous agents from teleoperated demonstrations? As motion is best seen in videos, we provide visual evaluations in our `website`.

### 4.1 Whole-body Motion Tracking

**Experiment Setup**. To answer **Q1**, we evaluate OmniH2O on motion tracking in simulation (Section 4.1.1) and the real world (Section 4.1.1). In simulation, we evaluate on the retargeted AMASS dataset with augmented motions $\hat{Q}$ (14k sequences); in real-world, we test on 20 standing sequences due to the limited physical lab space and the difficulty of evaluating on large-scale datasets in the real world. Detailed state-space composition (Appendix C), ablation setup (Appendix B), hyperparameters (Appendix K), and hardware configuration (Appendix A) are summarized in the Appendix.

**Metrics**. We evaluate the motion tracking performance using both pose and physics-based metrics. We report Success rate (Succ) as in PHC [56], where imitation is unsuccessful if the average deviation from reference is farther than 0.5m at any point in time. Succ measures whether the humanoid can track the reference motion without losing balance or lagging behind. The global MPJPE $E_{g-\text{mpjpe}}$ and the root-relative mean per-joint position error (MPJPE) $E_{\text{mpjpe}}$ (in mm) measures our policy's ability to imitate the reference motion globally and locally (root-relative). To show physical realism, we report average joint acceleration $E_{\text{acc}}$ (mm/frame$^2$) and velocity $E_{\text{vel}}$ (mm/frame) error.

#### 4.1.1 Simulation Motion-Tracking Results

In Table 1's first three rows, we can see that our deployable student policy significantly improves upon prior art [3] on motion imitation and achieves a similar success rate as the teacher policy.

**Ablation on DAgger/RL**. We test the performance of OmniH2O without DAgger (*i.e.*, directly using RL to train the student policy). In Table 1(a) we can see that DAgger improves performance

Table 1: Simulation motion imitation evaluation of OmniH2O and baselines on dataset $\hat{Q}$. Note that all the variants are trained with exact same rewards, domain randomizations and motion dataset $\hat{Q}$.

| Method | State Dimension | Sim2Real | All sequences | | | | | Successful sequences | | | |
|---|---|---|---|---|---|---|---|---|---|---|---|
| | | | Succ ↑ | $E_{\text{g-mpjpe}}$ ↓ | $E_{\text{mpjpe}}$ ↓ | $E_{\text{acc}}$ ↓ | $E_{\text{vel}}$ ↓ | $E_{\text{g-mpjpe}}$ ↓ | $E_{\text{mpjpe}}$ ↓ | $E_{\text{acc}}$ ↓ | $E_{\text{vel}}$ ↓ |
| Privileged policy | $\mathcal{S} \subset \mathcal{R}^{913}$ | ✗ | 94.77% | 126.51 | 70.68 | 3.57 | 6.20 | 122.71 | 69.06 | 2.22 | 5.20 |
| H2O [3] | $\mathcal{S} \subset \mathcal{R}^{138}$ | ✓ | 87.52% | 148.13 | 81.06 | 5.12 | 7.89 | 133.28 | 75.99 | 2.40 | 5.75 |
| OmniH2O | $\mathcal{S} \subset \mathcal{R}^{1665}$ | ✓ | **94.10%** | **141.11** | **77.82** | **3.70** | **6.54** | 135.49 | 75.75 | 2.30 | **5.47** |
| **(a) Ablation on DAgger/RL** | | | | | | | | | | | |
| OmniH2O-w/o-DAgger-History0 | $\mathcal{S} \subset \mathcal{R}^{90}$ | ✗ | 90.62% | 163.44 | 91.29 | 5.12 | 8.80 | 153.31 | 87.59 | 3.15 | 7.27 |
| OmniH2O-w/o-DAgger | $\mathcal{S} \subset \mathcal{R}^{1665}$ | ✗ | 47.11% | 223.27 | 128.90 | 15.03 | 16.29 | 182.13 | 119.54 | 5.47 | 9.10 |
| OmniH2O-History0 | $\mathcal{S} \subset \mathcal{R}^{90}$ | ✓ | 93.80% | 141.21 | 78.52 | 3.74 | 6.62 | **134.90** | 76.11 | **2.25** | 5.48 |
| OmniH2O | $\mathcal{S} \subset \mathcal{R}^{1665}$ | ✓ | **94.10%** | 141.11 | 77.82 | 3.70 | 6.54 | 135.49 | 75.75 | 2.30 | 5.47 |
| **(b) Ablation on History steps/Architecture** | | | | | | | | | | | |
| OmniH2O-History50 | $\mathcal{S} \subset \mathcal{R}^{3240}$ | ✓ | 93.56% | 141.51 | 78.51 | 4.01 | 6.79 | 135.04 | 76.07 | 2.36 | 5.55 |
| OmniH2O-History5 | $\mathcal{S} \subset \mathcal{R}^{405}$ | ✓ | 93.60% | **139.23** | **77.82** | 3.91 | 6.66 | 132.67 | 75.33 | 2.24 | 5.41 |
| OmniH2O-History0 | $\mathcal{S} \subset \mathcal{R}^{90}$ | ✓ | 93.80% | 141.21 | 78.52 | 3.74 | 6.62 | **134.90** | 76.11 | **2.25** | 5.48 |
| OmniH2O-GRU | $\mathcal{S} \subset \mathcal{R}^{90}$ | ✓ | 92.85% | 147.67 | 80.84 | 4.05 | 6.93 | 142.75 | 79.10 | 2.38 | 5.66 |
| OmniH2O-LSTM | $\mathcal{S} \subset \mathcal{R}^{90}$ | ✓ | 91.03% | 147.36 | 80.34 | 4.12 | 7.04 | 142.64 | 78.59 | 2.37 | 5.72 |
| OmniH2O-History25 (Ours) | $\mathcal{S} \subset \mathcal{R}^{1665}$ | ✓ | **94.10%** | 141.11 | 77.82 | 3.70 | 6.54 | 135.49 | 75.75 | 2.30 | **5.47** |
| **(c) Ablation on Tracking Points** | | | | | | | | | | | |
| OmniH2O-22points | $\mathcal{S} \subset \mathcal{R}^{1836}$ | ✓ | **94.72%** | **127.71** | **70.39** | **3.62** | **6.25** | **123.87** | **68.92** | **2.22** | **5.24** |
| OmniH2O-8points | $\mathcal{S} \subset \mathcal{R}^{1710}$ | ✓ | 94.31% | 129.30 | 71.70 | 3.78 | 6.39 | 125.14 | 70.07 | 2.22 | 5.26 |
| OmniH2O-3points (Ours) | $\mathcal{S} \subset \mathcal{R}^{1665}$ | ✓ | 94.10% | 141.11 | 77.82 | 3.70 | 6.54 | 135.49 | 75.75 | 2.30 | 5.47 |
| **(d) Ablation on Linear Velocity** | | | | | | | | | | | |
| OmniH2O-w-linvel | $\mathcal{S} \subset \mathcal{R}^{1743}$ | ✓ | 93.80% | **138.18** | 78.12 | 3.94 | 6.61 | **132.44** | 75.98 | **2.29** | **5.40** |
| OmniH2O | $\mathcal{S} \subset \mathcal{R}^{1665}$ | ✓ | **94.10%** | 141.11 | 77.82 | 3.70 | 6.54 | 135.49 | 75.75 | 2.30 | 5.47 |

overall, especially for policy with history input. Without DAgger the policy struggles to learn a coherent policy when provided with a long history. This is due to RL being unable to handle the exponential growth in input complexity. However, the history information is necessary for learning a deployable policy in the real-world, providing robustness and implicit global velocity information (see Section 4.1.2). Supervised learning via DAgger is able to effectively leverage the history input and is able to achieve better performance.

**Ablation on History Steps/Architecture**. In Table 1(b), we experiment with varying history steps (0, 5, 25, 50) and find that 25 steps achieve the best balance between performance and learning efficiency. Additionally, we evaluate different neural network architectures for history utilization: MLP, LSTM, GRU and determine that MLP-based OmniH2O performs the best.

**Ablation on Sparse Input**. To support VR-based teleoperation, $\pi_{\text{OmniH2O}}$ only tracks 3-points (head and hands) to produce whole-body motion. The impact of the number of tracking points is examined in Table 1(c). We test configurations ranging from minimal (3) to full-body motion goal (22) and found that 3-point tracking can achieve comparable performance with more input keypoints. As expected, 3-point policy sacrifices some whole-body motion tracking accuracy but gains greater applicability to commercially available devices.

**Ablation on Global Linear Velocity**. Given the challenges associated with global velocity estimation in real-world applications, we compare policies trained with and without explicit velocity information. In Table 1(d), we find that linear velocity information does not boost performance in simulation, but it introduces significant challenges in real-world deployment (details illustrated in Section 4.1.2), prompting us to develop a policy with state spaces that do not depend on linear velocity as proprioception to avoid these issues.

#### 4.1.2 Real-world Motion-Tracking Results

**Ablation on Real-world Linear Velocity Estimation.** We exclude linear velocity in our state space design global linear velocity obtained by algorithms such as visual inertial odometry (VIO) can be rather noisy, as shown in Appendix G. Our ablation study (Table 2(a)) also shows that policies without velocity input has better performance compared with policies using velocities estimated by VIO or MLP/GRU neural estimators (implementation details in

Table 2: Real-world motion tracking evaluation on 20 standing motions in $\hat{Q}$

| Method | State Dimensions | Tested sequences | | | |
|---|---|---|---|---|---|
| | | $E_{\text{g-mpjpe}}$ ↓ | $E_{\text{mpjpe}}$ ↓ | $E_{\text{acc}}$ ↓ | $E_{\text{vel}}$ ↓ |
| H2O [3] | $\mathcal{S} \subset \mathcal{R}^{138}$ | 87.33 | 53.32 | 6.03 | 5.87 |
| OmniH2O | $\mathcal{S} \subset \mathcal{R}^{1665}$ | **47.94** | **41.87** | **1.84** | **2.20** |
| **(a) Ablation on Real-world Linear Velocity Estimation** | | | | | |
| OmniH2O-w-linvel(VIO)[1,2] | $\mathcal{S} \subset \mathcal{R}^{1743}$ | N/A | N/A | N/A | N/A |
| OmniH2O-w-linvel(MLP) | $\mathcal{S} \subset \mathcal{R}^{1743}$ | 50.93 | 42.47 | 2.16 | 2.26 |
| OmniH2O-w-linvel(GRU) | $\mathcal{S} \subset \mathcal{R}^{1743}$ | 49.75 | 42.38 | 2.20 | 2.31 |
| OmniH2O | $\mathcal{S} \subset \mathcal{R}^{1665}$ | **47.94** | **41.87** | **1.84** | **2.20** |
| **(b) Ablation on History steps/Architecture** | | | | | |
| OmniH2O-History0 | $\mathcal{S} \subset \mathcal{R}^{90}$ | 83.26 | 46.00 | 4.86 | 4.45 |
| OmniH2O-History5 | $\mathcal{S} \subset \mathcal{R}^{405}$ | 62.18 | 46.50 | 2.66 | 2.90 |
| OmniH2O-History50 | $\mathcal{S} \subset \mathcal{R}^{3240}$ | 50.24 | **40.11** | 2.37 | 2.71 |
| OmniH2O-LSTM | $\mathcal{S} \subset \mathcal{R}^{90}$ | 87.00 | 46.06 | 3.89 | 3.88 |
| OmniH2O | $\mathcal{S} \subset \mathcal{R}^{1665}$ | **47.94** | 41.87 | **1.84** | **2.20** |

[1] Use ZED SDK to estimate the linear velocity.
[2] Unable to finish the real-world test due to falling on the ground.

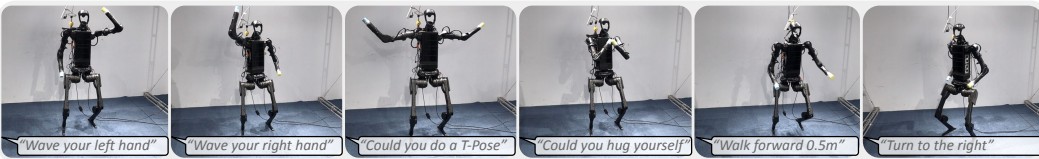

Figure 4: OmniH2O policy tracks motion goals from a language-based human motion generative model [57].

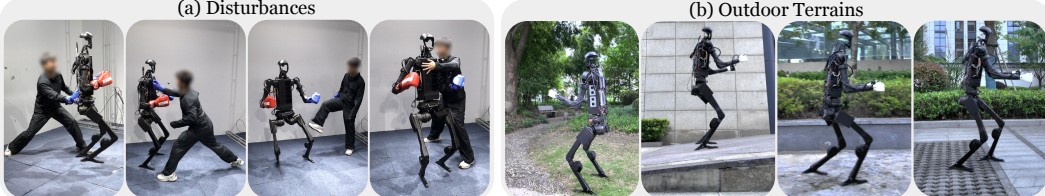

Figure 5: OmniH2O shows superior robustness against human strikes and different outdoor terrains.

Appendix G), which suggests that the policy with history can effectively track motions without explicit linear velocity as input.

**History Steps and Architecture**. Real-world evaluation in Table 2(b) also shows that our choice of 25 steps of history achieves the best performance. The tracking performance of LSTM shows that MLP-based policy performs better in the real-world.

## 4.2 Human Control via Universal Interfaces

To answer **Q2**, we demonstrate real-world capabilities of OmniH2O with versatile human control interfaces. All the capabilities discussed below utilize the same motion-tracking policy $\pi_{\text{OmniH2O}}$.

**Teleoperation**. We teleoperate the humanoid using $\pi_{\text{OmniH2O}}$ with both VR and RGB camera as interfaces. The results are shown in Figure 1(a) and Appendix I, where the robot is able to finish dexterous loco-manipulation tasks with high precision and robustness.

**Language Instruction Control**. By linking $\pi_{\text{OmniH2O}}$ with a pretrained text to motion generative model (MDM) [57], it enables controlling the humanoid via verbal instructions. As shown in Figure 4, with humans describing desired motions, such as "*raise your right hand*". MDM generates the corresponding motion goals that are tracked by the OmniH2O.

**Robustness Test**. As shown in Figure 5, we test the robustness of our control policy. We use the same policy $\pi_{\text{OmniH2O}}$ across all tests, whether with fixed standing motion goals or motion goals controlled by joysticks, either moving forward or backward. With human punching and kicking from various angles, the robot, without external assistance, is able to maintain stability on its own. We also test OmniH2O on various outdoor terrains, including grass, slopes, gravel, *etc*. OmniH2O demonstrates great robustness under disturbances and unstructured terrains.

## 4.3 Autonomy via Frontier Models or Imitation Learning

To answer **Q3**, we need to bridge the whole-body tracking policy (*physical intelligence*), with automated generation of kinematic motion goals through visual input (*semantic intelligence*). We explore two ways of automating humanoid control with OmniH2O: (1) using multi-modal frontier models to generate motion goals and (2) learning autonomous policies from the teleoperated dataset.

**GPT-4o Autonomous Control**. We integrate our system, OmniH2O, with GPT-4o, utilizing a head-mounted camera on the humanoid to capture images for GPT-4o (Figure 6). The prompt (details in Appendix M) provided to GPT-4o offers several motion primitives for it to choose from, based on the current visual context. We opt for motion primitives rather than directly generating motion goals because of GPT-4o's relatively long response time. As shown in Figure 6, the robot manages to give the correct punch based on the color of the target and successfully greets a human based on the intention indicated by human poses.

***OmniH2O-6* Dataset**. We collect demonstration data via VR-based teleoperation. We consider six tasks: Catch-Release, Squat, Rope-Paper-Scissors, Hammer-Catch, Boxing, and Pasket-Pick-Place. Our dataset includes paired RGBD images from the head-mounted camera, the motion goals

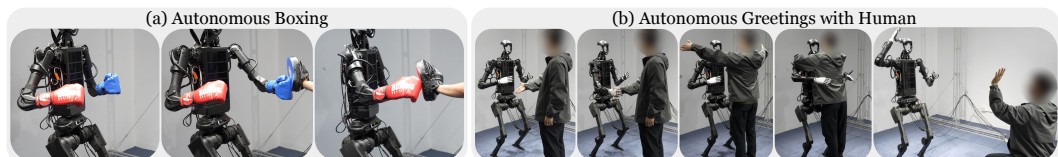

Figure 6: OmniH2O sends egocentric RGB views to GPT-4o and executes the selected motion primitives.

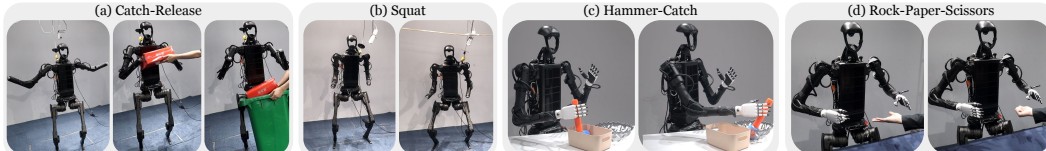

Figure 7: OmniH2O autonomously conducts four tasks using LfD models trained with our collected data.

of H1's head and hands with respect to the root, and joint targets for motor actuation, recorded at 30Hz. For simple tasks such as Catch-Release, Squat, and Rope-Paper-Scissors, approximately 5 minutes of data are recorded, and for tasks like Hammer-Catch and Basket-Pick-Place, we collect approximately 10 minutes, leading to 40-min real-world humanoid teleoperated demonstrations in total. Detailed task descriptions of the six open-sourced datasets are in Appendix J.

**Humanoid Learning from Demonstrations**. We design our learning from demonstration policy to be $\pi_{\text{LfD}}(\hat{\boldsymbol{p}}_{t:t+\phi}^{\text{Sparse-lfd}}|\boldsymbol{I}_t)$, where $\pi_{\text{LfD}}$ outputs $\phi$ frames of motion goals given the image input $\boldsymbol{I}_t$. Here, we also include dexterous hand commands in $\hat{\boldsymbol{p}}_{t:t+\phi}^{\text{Sparse-lfd}}$. Then, our $\pi_{\text{OmniH2O}}(\boldsymbol{a}_t|\boldsymbol{s}_t^{\text{p-real}}, \boldsymbol{s}_t^{\text{g-real}})$ serves as the low-level policy to compute joint actuations for humanoid whole-body control. The training hyperparameters are in Appendix L. Compared to directly using the $\pi_{\text{LfD}}$ to output joint actuation, we leverage the trained motor skills in $\pi_{\text{OmniH2O}}$, which drastically reduces the number of demonstrations needed. We benchmark a variety of imitation learning algorithms on four tasks in our collected dataset (shown in Figure 7), including Diffusion Policy [58] with Denoising Dif-

Table 3: Quantitative LfD average performance on 4 tasks over 10 runs.

| Metrics | All Tasks | | |
|---|---|---|---|
| **(a) Ablation on Data size** | | | |
| | 25%data | 50%data | 100%data |
| MSE Loss | 1.30E-2 | 7.48E-3 | 5.25E-4 |
| Succ rate | 4/10 | 6.5/10 | 8/10 |
| **(b) Ablation on Sequence observation/action** | | | |
| | Si-O-Si-A | Se-O-Se-A | Si-O-Se-A |
| MSE Loss | 4.89E-4 | 9.91E-4 | 5.25E-4 |
| Succ rate | 6.5/10 | 8.75/10 | 8/10 |
| **(c) Ablation on BC/DDIM/DDPM** | | | |
| | BC | DP-DDIM | DP-DDPM |
| MSE Loss | 5.63E-3 | 1.9E-3 | 5.25E-4 |
| Succ rate | 1/10 | 7.75/10 | 8/10 |

fusion Probabilistic Model [59] (DP-DDPM) and Denoising Diffusion Implicit Model [60] (DP-DDIM) and vanilla Behavior Cloning with a ResNet architecture (BC). Detailed descriptions of these methods are provided in Appendix D. To evaluate $\pi_{\text{LfD}}$, we report the average MSE loss and the success rate in Table 3, where we average the metrics across all tasks. More details for each task evaluation can be found in Appendix J. We draw two key conclusions: (1) The Diffusion Policy significantly outperforms vanilla BC with ResNet; (2) In our LfD training, predicting a sequence of actions is crucial, as it enables the robot to effectively learn and replicate the trajectory.

## 5 Limitations and Future Work

**Summary**. OmniH2O enables dexterous whole-body humanoid loco-manipulation via teleoperation, designs universal control interfaces, facilitates scalable demonstration collection, and empowers humanoid autonomy via frontier models or humanoid learning from demonstrations.

**Limitations**. One limitation of our system is the requirement of robot root odometry to transfer pose estimation from teleoperation interfaces to motion goals in the robot frame. Results from VIO can be noisy or even discontinuous, causing the motion goals to deviate from desired control. Another limitation is safety; although the OmniH2O policy has shown great robustness, we do not have guarantees or safety checks for extreme disturbances or out-of-distribution motion goals (*e.g.*, large discontinuity in motion goals). Future work could also focus on the design of the teleoperation system to allow the humanoid to traverse stairs with only sparse upper-body motion goals. Another interesting direction is improving humanoid learning from demonstrations by incorporating more sensors (*e.g.*, LiDAR, wrist cameras, tactile sensors) and better learning algorithms. We hope that our work spurs further efforts toward robust and scalable humanoid teleoperation and learning.

**Acknowledgments**

The authors express their gratitude to Ziwen Zhuang, Xuxin Cheng, Jiahang Cao, Wentao Dong, Toru Lin, Ziqiao Ma, Aoran Chen, Chunyun Wen, Unitree Robotics, Inspire Robotics, Damiao Technology for valuable help on the experiments and graphics design.

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

# Appendix

## Appendix

More real-world experiment videos are at the website <https://omni.human2humanoid.com>.

## A  Real Robot System Setup

Our real robot employs a Unitree H1 platform [61], outfitted with Damiao DM-J4310-2EC motors [62] and Inspire hands [63] for its manipulative capabilities. We have two versions of real robot computing setup. (1) The first one has two 16GB Orin NX computers mounted on the back of the H1 robot. The first Orin NX is connected to a ZED camera mounted on the waist of H1, which performs computations to determine H1's own location for positioning. The camera operates at 60 Hz FPS. Additionally, this Orin NX connects via Wi-Fi to our Vision Pro device to continuously receive motion goal information from a human operator. The second Orin NX serves as our main control hub. It receives the motion goal information, which it uses as input for our control policy. This policy then outputs torque information for each of the robot's motors and sends these commands to the robot. As control for the robot's fingers and wrists does not require inference, it is directly mapped from the Vision Pro data to the corresponding joints on the robot. The policy's computation frequency is set at 50 Hz. The two Orin NX units are connected via Ethernet, sharing information through a common ROS (Robot Operating System) network. The final commands to H1 are consolidated and dispatched by the second Orin NX. Our entire system has a low latency of only 20 milliseconds. It's worth noting that we designed the system in this way partly because the ZED camera requires

substantial computational resources. By dedicating the first Orin NX to the ZED camera, and the second to policy inference, we ensure that each component operates with optimal performance. (2) In the second setup, a laptop (13th Gen i9-13900HX and NVIDIA RTX4090, 32GB RAM) serves as the computing and communication device. All devices, including the ZED camera, control policy, and Vision Pro, communicate through this laptop on its ROS system, facilitating centralized data handling and command dispatch. These two setups yield similar performance, and we use them interchangeably in our experiments.

## B   Simulation Baseline and Ablations

In this section, we provide an explanation of each ablation method.  **Main results in Table 1**

- **Privileged policy**: This teacher policy $\pi_{\text{privileged}}$ incorporates all privileged environment information, along with complete motion goal and proprioception data in the observations. State space composition details in Table 4.

- **H2O**: A policy trained using RL without DAgger and historical data, utilizing 8 keypoints of motion goal in observations. State space composition details in Table 5.

- **OmniH2O**: Our deployment policy $\pi_{\text{OmniH2O}}$ that includes historical information and uses 3 keypoints of motion goal in observations, trained with DAgger. State space composition details in Table 6.

**Ablation on DAgger/RL in Table 1(a)**

- **OmniH2O-w/o-DAgger-History0**: This variant of OmniH2O is trained solely using RL and does not incorporate historical information within observations. State space composition details in Table 7.

- **OmniH2O-w/o-DAgger**: This model is trained using RL, excludes DAgger, but includes historical information from the last 25 steps in observations. State space composition details in Table 8.

- **OmniH2O-History0**: This model is trained with DAgger, but excludes historical information from the last 25 steps in observations. State space composition details in Table 9.

- **OmniH2O**: This model is trained with DAgger and incorporates 25-step historical information within observations. State space composition details in Table 6.

**Ablation on History steps/Architecture in Table 1(b)**

- **OmniH2O-History50/25/5/0**: This variant of the OmniH2O with 50, 25, or 0 steps of historical information in the observations. State space composition details in Table 10.

- **OmniH2O-GRU/LSTM**: This version replaces the MLP in the policy network with either GRU or LSTM, inherently incorporating historical observations. State space composition details in Table 11.

**Ablation on Tracking Points in Table 1(c)**

- **OmniH2O-22/8/3points**: This variant of the OmniH2O policy includes 22, 8, or 3 keypoints of motion goal in the observations, with the 3 keypoints setting corresponding to the standard OmniH2O policy. State space composition details in Tables 6, 12 and 13.

**Ablation on Linear Velocity in Table 1(d)**

- **OmniH2O-w-linvel**: This variant is almost the same as OmniH2O but with root linear velocity in observations and past linear velocity in history information. State space composition details in Table 14.

# C   State Space Compositions

In this section, we introduce the detailed state space composition of baselines in the experiments.

**Privileged Policy.** This policy $\pi_{\text{privileged}}$ is the teacher policy that has access to all the available states for motion imitation, trained using RL.

Table 4: State space information in Privileged Policy setting

| State term | Dimensions |
|---|---|
| (Proprioception) Rigid Body position | 66 |
| (Proprioception) Rigid Body rotation | 138 |
| (Proprioception) Rigid Body velocity | 69 |
| (Proprioception) Rigid Body angular velocity | 69 |
| (Motion goal) Rigid Body position difference | 69 |
| (Motion goal) Rigid Body rotation difference | 138 |
| (Motion goal) Rigid Body velocity difference | 69 |
| (Motion goal) Rigid Body angular velocity difference | 69 |
| (Motion goal) Local Rigid Body position | 69 |
| (Motion goal) Local Rigid Body rotation | 138 |
| Actions | 19 |
| Total dim | 913 |

**H2O.** This policy has 8 keypoints input (shoulder, elbow, hand, leg) and with global linear velocity, trained using RL.

Table 5: State space information in H2O setting

| State term | Dimensions |
|---|---|
| DoF position | 19 |
| DoF velocity | 19 |
| Base velocity | 3 |
| Base angular velocity | 3 |
| Base gravity | 3 |
| Motion goal | 72 |
| Actions | 19 |
| Total dim | 138 |

**OmniH2O.** This is our deployment policy $\pi_{\text{OmniH2O}}$ with 25 history steps and without global linear velocity, trained using DAgger.

Table 6: State space information in OmniH2O setting

| State term | Dimensions |
| --- | --- |
| DoF position | 19 |
| DoF velocity | 19 |
| Base angular velocity | 3 |
| Base gravity | 3 |
| Motion goal | 27 |
| Actions | 19 |
| Single step total dim | 90 |
| History state term | Dimensions |
| DoF position | 19 |
| DoF velocity | 19 |
| Base angular velocity | 3 |
| Base gravity | 3 |
| Actions | 19 |
| History Single step total dim | 63 |
| Total dim | 1665(63*25 + 90) |

**OmniH2O-w/o-DAgger-History0.** This policy has a history of 0 steps, trained using RL.

Table 7: State space information in OmniH2O-w/o-DAgger-History0 setting

| State term | Dimensions |
| --- | --- |
| DoF position | 19 |
| DoF velocity | 19 |
| Base angular velocity | 3 |
| Base gravity | 3 |
| Motion goal | 27 |
| Actions | 19 |
| Total dim | 90 |

**OmniH2O-w/o-DAgger.** This policy has the same architecture as OmniH2O but trained with RL.

Table 8: State space information in OmniH2O-w/o-DAgger setting

| State term | Dimensions |
| --- | --- |
| DoF position | 19 |
| DoF velocity | 19 |
| Base angular velocity | 3 |
| Base gravity | 3 |
| Motion goal | 27 |
| Actions | 19 |
| Single step total dim | 90 |
| History state term | Dimensions |
| DoF position | 19 |
| DoF velocity | 19 |
| Base angular velocity | 3 |
| Base gravity | 3 |
| Actions | 19 |
| History Single step total dim | 63 |
| Total dim | 1665(63*25 + 90) |

**OmniH2O-History0.** This policy has a history of 0 steps, trained using DAgger.

Table 9: State space information in OmniH2O-History0 setting

| State term | Dimensions |
| --- | --- |
| DoF position | 19 |
| DoF velocity | 19 |
| Base angular velocity | 3 |
| Base gravity | 3 |
| Motion goal | 27 |
| Actions | 19 |
| Total dim | 90 |

**OmniH2O-History*x*.** This policy has a history of *x* steps, trained using DAgger.

Table 10: State space information in OmniH2O-History*x* setting

| State term | Dimensions |
| --- | --- |
| DoF position | 19 |
| DoF velocity | 19 |
| Base angular velocity | 3 |
| Base gravity | 3 |
| Motion goal | 27 |
| Actions | 19 |
| Single step total dim | 90 |
| History state term | Dimensions |
| DoF position | 19 |
| DoF velocity | 19 |
| Base angular velocity | 3 |
| Base gravity | 3 |
| Actions | 19 |
| History Single step total dim | 63 |
| Total dim | 63*$x$ + 90 |

**OmniH2O-GRU/LSTM.** This policy uses GRU/LSTM-based architecture, trained using DAgger.

Table 11: State space information in OmniH2O-GRU/LSTM setting

| State term | Dimensions |
| --- | --- |
| DoF position | 19 |
| DoF velocity | 19 |
| Base angular velocity | 3 |
| Base gravity | 3 |
| Motion goal | 27 |
| Actions | 19 |
| Total dim | 90 |

**OmniH2O-22points.** This policy has 22 keypoints input (every joint on the humanoid), trained using DAgger.

Table 12: State space information in OmniH2O-22points setting

| State term | Dimensions |
|---|---|
| DoF position | 19 |
| DoF velocity | 19 |
| Base angular velocity | 3 |
| Base gravity | 3 |
| Motion goal | 198 |
| Actions | 19 |
| Single step total dim | 261 |
| History state term | Dimensions |
| DoF position | 19 |
| DoF velocity | 19 |
| Base angular velocity | 3 |
| Base gravity | 3 |
| Actions | 19 |
| History Single step total dim | 63 |
| Total dim | 1836(63*25+261 ) |

**OmniH2O-8points.** This policy has 8 keypoints input (shoulder, elbow, hand, leg), trained using DAgger.

Table 13: State space information in OmniH2O-8points setting

| State term | Dimensions |
|---|---|
| DoF position | 19 |
| DoF velocity | 19 |
| Base angular velocity | 3 |
| Base gravity | 3 |
| Motion goal | 72 |
| Actions | 19 |
| Single step total dim | 135 |
| History state term | Dimensions |
| DoF position | 19 |
| DoF velocity | 19 |
| Base angular velocity | 3 |
| Base gravity | 3 |
| Actions | 19 |
| History Single step total dim | 63 |
| Total dim | 1710(63*25+135 ) |

**OmniH2O-w-linvel.** This policy has 25 history steps and global linear velocity, trained using DAgger.

Table 14: State space information in OmniH2O-w-linvel setting

| State term | Dimensions |
|---|---|
| DoF position | 19 |
| DoF velocity | 19 |
| Base velocity | 3 |
| Base angular velocity | 3 |
| Base gravity | 3 |
| Motion goal | 27 |
| Actions | 19 |
| Single step total dim | 93 |
| History state term | Dimensions |
| DoF position | 19 |
| DoF velocity | 19 |
| Base velocity | 3 |
| Base angular velocity | 3 |
| Base gravity | 3 |
| Actions | 19 |
| History Single step total dim | 66 |
| Total dim | 1743(66*25 + 93) |

## D  LfD Baselines

We conduct numerous ablation studies on LfD, aiming to benchmark the impact of various aspects on LfD tasks. The details of each ablation are as follows:

**Ablation on Dataset size**.

- 25/50/100% data: In this task, we use 25/50/100% of the dataset as the training set. The algorithm is DDPM which takes a single-step image as input and outputs 8 steps of actions.

**Ablation on Single/Sequence observation/action input/output**.

- Si-O-Si-A: Single-step observation and single-step action mean that we take 1 step of image data as input and predict 1 step of action as output.

- Se-O-Se-A: Sequence-steps observation and sequence-steps actions mean that we take 4 steps of image data as input and predict 8 steps of action as output.

- Si-O-Se-A: Single-step observation and sequence-steps actions mean that we take 1 step of image data as input and predict 8 steps of action as output.

**Ablation on Training Architecture.**.

- BC: Behavior cloning which means we use resnet+MLP to predict the next 8 steps action from the current step's image.

- DP-DDIM: We use DDIM as the algorithm which takes a single-step image as input and outputs 8 steps of actions.

- DP-DDPM: We use DDPM as the algorithm which takes a single-step image as input and outputs 8 steps of actions.

## E  Reward Functions

**Reward Components**. Detailed reward components are summarized in Table 15.

Table 15: Reward components and weights: penalty rewards for preventing undesired behaviors for sim-to-real transfer, regularization to refine motion, and task reward to achieve successful whole-body tracking in real-time.

| Term | Expression | Weight |
|---|---|---|
| | Penalty | |
| Torque limits | $\mathbb{1}(\boldsymbol{\tau}_t \notin [\boldsymbol{\tau}_{\min}, \boldsymbol{\tau}_{\max}])$ | -2 |
| DoF position limits | $\mathbb{1}(\boldsymbol{d}_t \notin [\boldsymbol{q}_{\min}, \boldsymbol{q}_{\max}])$ | -125 |
| DoF velocity limits | $\mathbb{1}(\dot{\boldsymbol{d}}_t \notin [\dot{\boldsymbol{q}}_{\min}, \dot{\boldsymbol{q}}_{\max}])$ | -50 |
| Termination | $\mathbb{1}_{\text{termination}}$ | -250 |
| | Regularization | |
| DoF acceleration | $\|\ddot{\boldsymbol{d}}_t\|_E 2$ | -0.000011 |
| DoF velocity | $\|\dot{\boldsymbol{d}}_t\|_2^2$ | -0.004 |
| Lower-body action rate | $\|\boldsymbol{a}_t^{\text{lower}} - \boldsymbol{a}_{t-1}^{\text{lower}}\|_2^2$ | -3 |
| Upper-body action rate | $\|\boldsymbol{a}_t^{\text{upper}} - \boldsymbol{a}_{t-1}^{\text{upper}}\|_2^2$ | -0.625 |
| Torque | $\|\boldsymbol{\tau}_t\|$ | -0.0001 |
| Feet air time | $T_{\text{air}} - 0.25$ [64] | 1000 |
| Max feet height for each step | $\max\{\boldsymbol{h}_{\text{max feet height for each step}} - 0.25, 0\}$ | 1000 |
| Feet contact force | $\|F_{\text{feet}}\|_2^2$ | -0.75 |
| Stumble | $\mathbb{1}(F_{\text{feet}}^{xy} > 5 \times F_{\text{feet}}^z)$ | -0.00125 |
| Slippage | $\|\boldsymbol{v}_t^{\text{feet}}\|_2^2 \times \mathbb{1}(F_{\text{feet}} \geq 1)$ | -37.5 |
| Feet orientation | $\|\boldsymbol{g}_z^{\text{feet}}\|$ | -62.5 |
| In the air | $\mathbb{1}(F_{\text{feet}}^{\text{left}}, F_{\text{feet}}^{\text{right}} < 1)$ | -200 |
| Orientation | $\|\boldsymbol{g}_z^{\text{root}}\|$ | -200 |
| | Task Reward | |
| DoF position | $\exp(-0.25\|\hat{\boldsymbol{d}}_t - \boldsymbol{d}_t\|_2)$ | 32 |
| DoF velocity | $\exp(-0.25\|\dot{\hat{\boldsymbol{d}}}_t - \dot{\boldsymbol{d}}_t\|_2^2)$ | 16 |
| Body position | $\exp(-0.5\|\boldsymbol{p}_t - \hat{\boldsymbol{p}}_t\|_2^2)$ | 30 |
| Body position VRpoints | $\exp(-0.5\|\boldsymbol{p}_t^{\text{real}} - \hat{\boldsymbol{p}}_t^{\text{real}}\|_2^2)$ | 50 |
| Body rotation | $\exp(-0.1\|\boldsymbol{\theta}_t \ominus \hat{\boldsymbol{\theta}}_t\|)$ | 20 |
| Body velocity | $\exp(-10.0\|\boldsymbol{v}_t - \hat{\boldsymbol{v}}_t\|_2)$ | 8 |
| Body angular velocity | $\exp(-0.01\|\boldsymbol{\omega}_t - \hat{\boldsymbol{\omega}}_t\|_2)$ | 8 |

**Reward Curriculum**. We have modified the cumulative discounted reward expression to handle multiple small rewards at each time step differently, depending on their sign. The revised formula is given by: $\mathbb{E}\left[\sum_{t=1}^{T} \gamma^{t-1} \sum_i s_{t,i} r_{t,i}\right]$ where $r_{t,i}$ represents different reward functions at time $t$, and $s_{t,i}$ is the scaling factor for each reward, defined as: $s_{t,i} = \begin{cases} s_{\text{current}} & \text{if } r_{t,i} < 0 \\ 1 & \text{if } r_{t,i} \geq 0 \end{cases}$ where $s_{current}$ is the scaling factor. This scaling factor is adjusted dynamically: it is multiplied by 0.9999 when the average episode length is less than 40, and multiplied by 1.0001 when it exceeds 120. The init $s_{current}$ is set to 0.5, then the upper bound of this scaling factor is set to 1. This modification allows our policy to progressively learn from simpler to more complex scenarios with higher penalties, thereby reducing the difficulty for RL in exploring the optimal policy.

# F   Domain Randomizations

Detailed domain randomization setups are summarized in Table 16.

Table 16: Here we describe the range of dynamics randomization for simulated dynamics randomization, external perturbation, and terrain, which are important for sim-to-real transfer, robustness, and generalizability.

| Term | Value |
|---|---|
| **Dynamics Randomization** | |
| Friction | $\mathcal{U}(0.2, 1.1)$ |
| Base CoM offset | $\mathcal{U}(-0.1, 0.1)$m |
| Link mass | $\mathcal{U}(0.7, 1.3) \times$ default kg |
| P Gain | $\mathcal{U}(0.75, 1.25) \times$ default |
| D Gain | $\mathcal{U}(0.75, 1.25) \times$ default |
| Torque RFI [65] | $0.1 \times$ torque limit $N \cdot m$ |
| Control delay | $\mathcal{U}(20, 60)$ms |
| Motion reference offset | $\mathcal{U}([-0.02, 0.02], [-0.02, 0.02], [-0.1, 0.1])$cm |
| **External Perturbation** | |
| Push robot | interval $= 5s$, $v_{xy} = 1$m/s |
| **Randomized Terrain** | |
| Terrain type | flat, rough, low obstacles [20] |

## G   Linear Velocity Estimation

The illustration of using the ZED camera VIO module and the comparison of VIO with neural state estimators are shown in Figure 8. We train our neural velocity estimators using a supervised learning approach. The process involves repeatedly deploying our policy in simulation with different motion goals. In every environment step, we use the root linear velocity to supervise our velocity estimator.

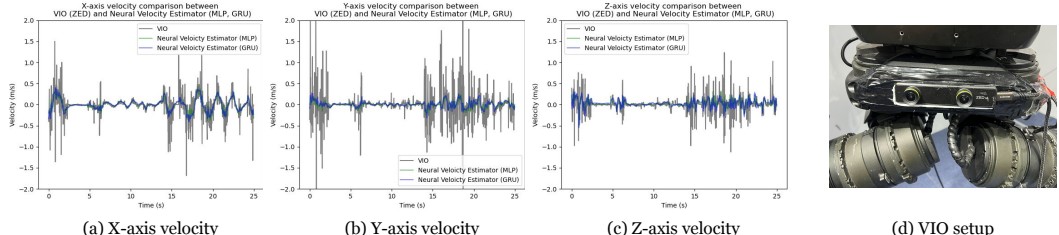

(a) X-axis velocity      (b) Y-axis velocity      (c) Z-axis velocity      (d) VIO setup

Figure 8: The illustration of using ZED camera VIO module, and the comparison of the velocity estimation of VIO with neural state estimators.

## H   Ablation on Dataset Motion Distribution

The ablation study on motion data distribution is shown in Figure 9. The policy trained without motion data augmentation is hard to stand still and make upper-body moves.

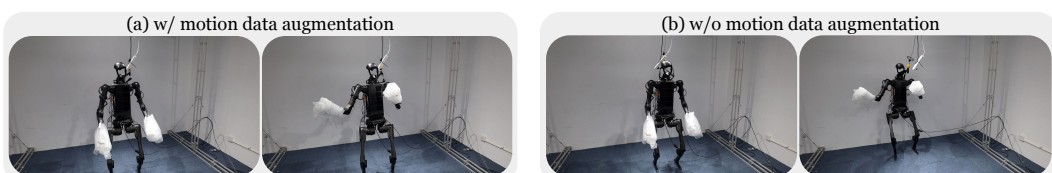

(a) w/ motion data augmentation      (b) w/o motion data augmentation

Figure 9: The ablation of data augmentation.

## I   Additional Physical Teleoperation Results

Additional VR-based and RGB-based teleoperation demo are shown in Figure 10.

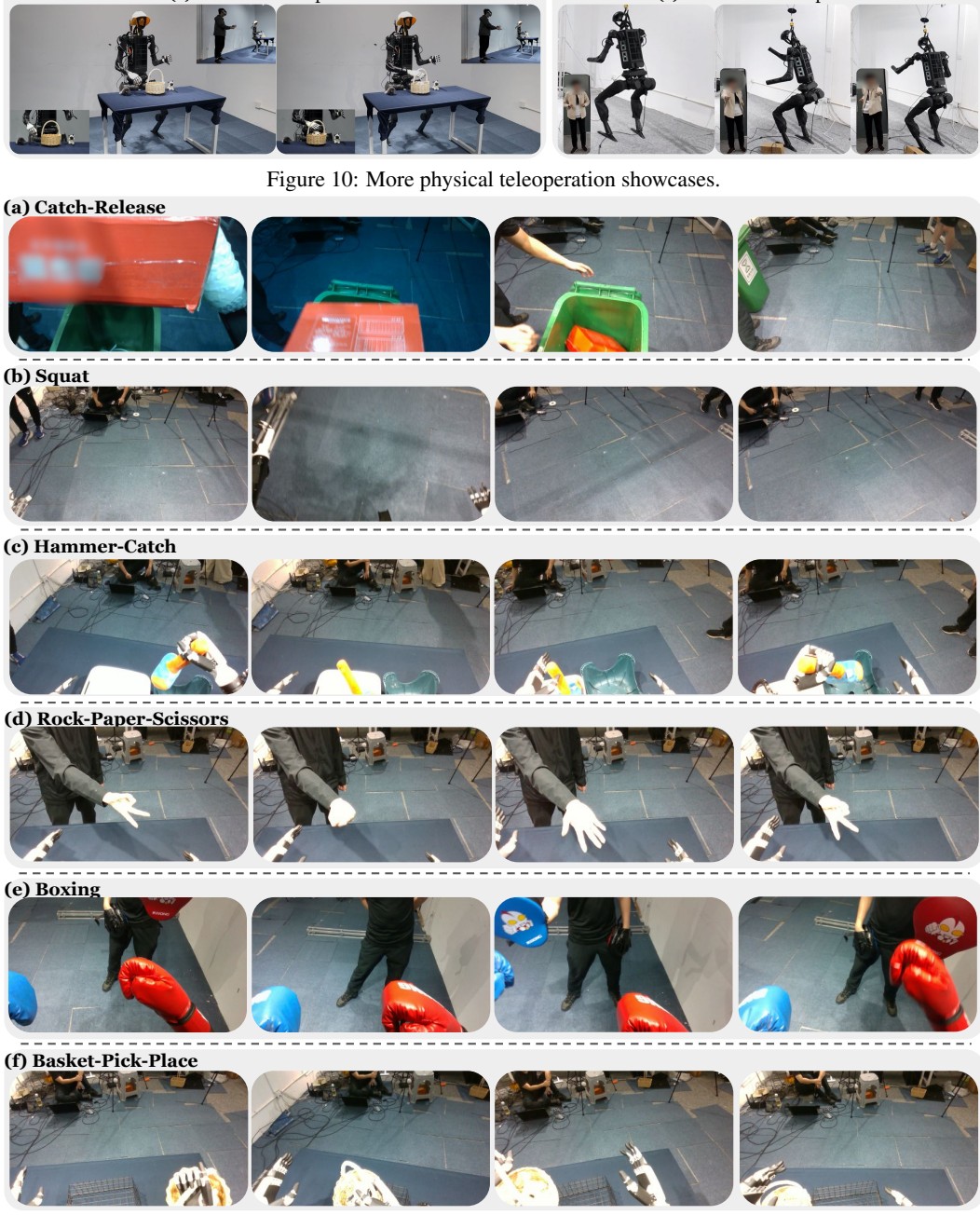

Figure 10: More physical teleoperation showcases.

Figure 11: *OmniH2O-6* dataset.

## J Dataset and Imitation Learning

As shown in Figure 11, we collected 6 LfD tasks' dataset to enable the robot to autonomously perform certain functions.

**Catch-Release**: Catch a red box and release it into a trash bin. This task has 13234 frames in total.

**Squat**: Squat when the robot sees a horizontal bar approaching that is lower than its head height. This task has 8535 frames in total.

**Hammer-Catch**: Use right hand to catch a hammer in a box. This task has 12759 frames in total.

**Rock-Paper-Scissors**: When the robot sees the person opposite it makes one of the rock-paper-scissors gestures, it should respond with the corresponding gesture that wins. This task has 9380 frames in total.

**Boxing**: When you see a blue boxing target, throw a left punch; when you see a red one, throw a right punch. This task has 11118 frames in total.

**Basket-Pick-Place**: Use your right hand to pick up the box and place it in the middle when the box is on the right side, and use your left hand if the box is on the left side. If you pick up the box with your right hand, place it on the left side using your left hand; if picked up with your left hand, place it on the right side using your right hand. This task has 18436 frames in total.

The detailed performance of 4 tasks is documented in Table 17

Table 17: Quantitative LfD autonomous agents performance for 4 tasks.

| Metrics | Catch-Release | | | Squat | | | Hammer-Catch | | | Rock-Paper-Scissors | | |
|---|---|---|---|---|---|---|---|---|---|---|---|---|
| **(a) Ablation on Data size** | | | | | | | | | | | | |
| | 25%data | 50%data | 100%data | 25%data | 50%data | 100%data | 25%data | 50%data | 100%data | 25%data | 50%data | 100%data |
| MSE Loss | 3.01E-3 | 3.04E-4 | 9.89E-5 | 1.25E-4 | 1.10E-4 | 7.07E-5 | 2.18E-2 | 1.56E-2 | 3.29E-4 | 2.72E-2 | 1.39E-2 | 1.60E-3 |
| Succ rate | 1/10 | 3/10 | 6/10 | 9/10 | 10/10 | 10/10 | 3/10 | 6/10 | 6/10 | 3/10 | 9/10 | 10/10 |
| **(b) Ablation on Sequence observation/action** | | | | | | | | | | | | |
| | Si-O-Si-A | Se-O-Se-A | Si-O-Se-A | Si-O-Si-A | Se-O-Se-A | Si-O-Se-A | Si-O-Si-A | Se-O-Se-A | Si-O-Se-A | Si-O-Si-A | Se-O-Se-A | Si-O-Se-A |
| MSE Loss | 2.52E-4 | 1.47E-4 | 9.89E-5 | 5.18E-5 | 9.60E-5 | 7.07E-5 | 2.22E-4 | 3.62E-4 | 3.29E-4 | 1.43E-3 | 3.36E-3 | 1.60E-3 |
| Succ rate | 3/10 | 7/10 | 6/10 | 10/10 | 10/10 | 10/10 | 5/10 | 9/10 | 6/10 | 10/10 | 9/10 | 10/10 |
| **(c) Ablation on BC/DDIM/DDPM** | | | | | | | | | | | | |
| | BC | DP-DDIM | DP-DDPM | BC | DP-DDIM | DP-DDPM | BC | DP-DDIM | DP-DDPM | BC | DP-DDIM | DP-DDPM |
| MSE Loss | 1.39E-3 | 4.79E-5 | 9.89E-5 | 6.24E-4 | 6.42E-5 | 7.07E-5 | 4.50E-3 | 3.41E-4 | 3.29E-4 | 1.46E-2 | 2.42E-3 | 1.60E-3 |
| Succ rate | 0/10 | 6/10 | 6/10 | 3/10 | 10/10 | 10/10 | 0/10 | 5/10 | 6/10 | 1/10 | 10/10 | 10/10 |

# K   Sim2real Training Hyperparameters

The hyperparameters for our RL/DAgger policy training are detailed in Table 18 below.

Table 18: Hyperparameters

| Hyperparameters | Values |
|---|---|
| Batch size | 64 |
| Discount factor ($\gamma$) | 0.99 |
| Learning rate | 0.001 |
| Clip param | 0.2 |
| Entropy coef | 0.005 |
| Max grad norm | 0.2 |
| Value loss coef | 1 |
| Entropy coef | 0.005 |
| Init noise std (RL) | 1.0 |
| Init noise std (DAgger) | 0.001 |
| Num learning epochs | 5 |
| MLP size | [512, 256, 128] |

# L   LfD Hyperparameters

In order to make the robot autonomous, we have developed a Learning from Demonstration (LfD) approach utilizing a diffusion policy that learns from a dataset we collected. The default training hyperparameters are shown below in Table 19.

Table 19: Training Hyperparameters for the Lfd Training

| Hyperparameter | Default Value |
| --- | --- |
| Batch Size | 32 |
| Observation Horizon | 1 |
| Action Horizon | 8 |
| Prediction Horizon | 16 |
| Policy Dropout Rate | 0.0 |
| Dropout Rate (State Encoder) | 0.0 |
| Image Dropout Rate | 0.0 |
| Weight Decay | 1E-5 |
| Image Output Size | 32 |
| State Noise | 0.0 |
| Image Gaussian Noise | 0.0 |
| Image Masking Probability | 0.0 |
| Image Patch Size | 16 |
| Number of Diffusion Iterations | 100 |

# M   GPT-4o Prompt Example

Here is the example prompt we use for **Autonomous Boxing** task:

*You're a humanoid robot equipped with a camera slightly tilted downward on your head, providing a first-person perspective. I am assigning you a task: when a blue target appears in front of you, extend and then retract your left fist. When a red target appears, do the same with your right fist. If there is no target in front, remain stationary. I will provide you with three options each time: move your left hand forward, move your right hand forward, or stay motionless. You should directly respond with the corresponding options A, B, or C based on the current image. Note that, yourself is also wearing blue left boxing glove and right red boxing glove, please do not recognize them as the boxing target. Now, based on the current image, please provide me with the A, B, C answers.*

For **Autonomous Greetings with Human** Task, our prompt is:

*You are a humanoid robot equipped with a camera slightly tilted downward on your head, providing a first-person perspective. I am assigning you a new task to respond to human gestures in front of you. Remember, the person is standing facing you, so be mindful of their gestures. If the person extends their right hand to shake hands with you, use your right hand to shake their right hand (Option A). If the person opens both arms wide for a hug, open your arms wide to reciprocate the hug (Option B). If you see the person waving his hand as a gesture to say goodbye, respond by waving back (Option C). If no significant gestures are made, remain stationary (Option D). Respond directly with the corresponding options A, B, C, or D based on the current image and observed gestures. Directly reply with A, B, C, or D only, without any additional characters.*

It is worth mentioning that we can use GPT-4 not only to choose motion primitive but also to directly generate the motion goal. The following prompt exemplifies this process:

*You are a humanoid robot equipped with a camera slightly tilted downward on your head, providing a first-person perspective. I am assigning you a new task to respond to human gestures in front of you. If the person extends his left hand for a handshake, extend your left hand to reciprocate. If they extend their right hand, respond by extending your right hand. If the person opens both arms wide for a hug, open your arms wide to reciprocate the hug. If no significant gestures are made, remain stationary. Respond 6 numbers to represent the desired left and right hand 3D position with respect to your root position. For example: [0.25, 0.2, 0.3, 0.15, -0.19, 0.27] means the desired position of the left hand is 0.25m forward, 0.2m left, and 0.3m high compared to pelvis position, and the desired position of the right hand is 0.15m forward, 0.19m right and 0.27m high compared to pelvis position. The default stationary position should be (0.2, 0.2, 0.2, 0.2, -0.2, 0.2). Now please respond the 6d array based on the image to respond to the right hand shaking, left hand shaking, and hugging.*

