# OpenReview forum: "OmniH2O: Universal and Dexterous Human-to-Humanoid Whole-Body Teleoperation and Learning"
_robot-learning.org/CoRL/2024/Conference — CoRL 2024_

### Official Review · Reviewer_huqu · 2024-07-15
**Interesting application of general humanoid teleoperation policy, but lack of advancement compared with previous works.**

**Originality:** 3
**Technical Quality:** 3
**Clarity Of Presentation:** 4
**Potential Impact:** 2
**Recommendation:** 3
**Confidence:** 5

**Review:**

This paper describes the bi-level framework of humanoid policy that can be commanded by vocal instructions. The writing is clear and the technical details are well-organized. The author provides a clear motivation. However, compared with previous works, the advancement of this work is not addressed clearly. The author should provide a clear comparison with previous works or address the advancement of this work with more detailed information.

**strength**

* The author provides a clear motivation and well-organized technical details.

* The author provides a clear description of the training framework of the teleoperation policy.

* Clear illustration on using GPT-4o for the full-autonomous tasks.

**weakness**

* The research advancement is not clear compared with previous works.

* Details of learning from demonstrations are not sufficient in the manuscript.

* Lack of thorough investigation on the performance of different methods in learning from demonstrations.

* Lack of details on the OmniH2O-6 dataset. For example, how is the lower-limb motion captured when collecting the dataset using VR? In the training section of the teleoperation policy, the policy is designed to follow the lower-limbs motion.

**Quality Of The Limitations Section:**

2

**Questions For Rebuttal:**

1. What is the reference trajectory when performing outdoor locomotion tests?

2. Why distillation is needed for teleoperation policy?

3. Is it possible to use a more accurate odometry system to provide the linear velocity of the robot to achieve a better performance? For example, using an in-door tracking system for a proof of concept.

**Robotics Focus:**

4

**Summary Of Paper:**

This paper introduces a bi-level autonomous humanoid policy that can be commanded by vocal instructions. The low-level policy can be operated by human operator by teleoperation. The high-level policy can be deployed using GPT-4o, MGM, or a trained diffusion policy. The author demonstrates the training framework of the low-level policy and releases the dataset OmniH2O-6 dataset to train the high-level diffusion policy.

**Summary Of Recommendation:**

The advancement of this work is not clear compared with previous works. I recommend a weak reject on this paper.

---

### Official Review · Reviewer_ziut · 2024-07-19
**Interesting and novel humanoid robotics sim2real demo with weaknesses in presentation and evaluation.**

**Originality:** 3
**Technical Quality:** 3
**Clarity Of Presentation:** 3
**Potential Impact:** 3
**Recommendation:** 3
**Confidence:** 4

**Review:**

Overall this is one of the first (among a couple others) works on sim2real learning of full-body humanoid mimicry. Thus, from an application perspective the novelty is high.

While I enjoyed reading the paper, I found some of the technical descriptions to be difficult to understand. In addition, there are certain weaknesses in the empirical investigation that can be improved. Below I mention some of the key issues I identified.

How did you select the squatting and standing pose used in the training? This could bias the system to less stable poses, which should ultimately depend on the task. More discussion of this choice would be useful.

The main paper should provide at least some description of the reward term "max feet height for each step." since that is claimed to be crucial and is not self-explanatory. I tried to parse Appendix E that the authors pointed to, but it mostly provided factual details without motivation or definition of the symbols.

I was not able to understand the rationale for the dynamic reward function described in the appendix. Is there a fundamental principle here that can be formalized more soundly?

Why didn't you try to include the hand control in the overall learning framework? The videos showing hand movement are a bit misleading and this should be made clear early on in the paper.

It would have been very interested to have at least a couple of non-standing real-world sequences. The additional dynamics of such sequences are non-trivial to deal with and it is important to know if the current approach has any fundamental limitations in that respect. Even a negative result would be interesting. The excuse given for not including such sequences (not enough lab space) is not very reasonable given videos already shown.

Many of the values in Table 1 are quite similar, including those that are marked in bold as somehow significantly better. I'm not sure I can agree with conclusions regarding the relative performance of some of these choices. These are also hard to evaluate since they are apparently averaged over all 14,000 sequences.  Was there any randomization in the experiments (e.g. multiple randomizations for each sequence)? If so, the confidence intervals could be given?

The OmniH20 NN architecture was not mentioned until the experimental section. This should be mentioned earlier.

I did not understand the "ablation to sparse input" section. Does your OmniH20 architecture allow inputs with different types of sparsity or do you need to train a student for each sparsity you consider. I tried to figure this out from the paper text, but couldn't piece things together.

How were the real-world experiments conducted? Was there just one trial of each motion? Why isn't there a success rate here?

How does Figure 5 show "superior robustness"? Compared to what? There is nothing quantitative here upon which to draw a conclusion.

Post Rebuttal:

I read the rebuttal to my review an the other reviews. I can only say that while I still appreciate the work, because there is very little that has been in this area so far, I now appreciate the limitations more.

I feel that the submission is overselling the demonstrated capabilities of this approach in a number of ways consistent with my questions. The submitted video of simulated dynamic motions to one of the other reviewers was not very convincing --- it clearly would not transfer to hardware. The video submitted of the 20 real-world tests was a bit of a mystery to me. Maybe that was just one motion.

I would say that the authors need to be much more upfront from the start about the lack of actual dynamic motion demonstrations, which are a major challenge. Otherwise, it is very easy for people to get a false sense of the state-of-the-art and actually not recognize future innovations when they occur---think this to be a solved problem.

Accordingly, I downgraded my ranking to a "weak accept".

**Quality Of The Limitations Section:**

2

**Questions For Rebuttal:**

Is there a fundamental principle regarding the non-stationary reward formulation described in the appendix?

Why didn't you try to include the hand control in the overall learning framework?

What can you say about the statistical significance of results in Table 1 (see comment in review)?

How does your approach deal with different levels of sparsity in the input ("ablation to sparse input")?

How were the real-world experiments conducted? Why isn't there a success rate here?

**Robotics Focus:**

4

**Summary Of Paper:**

The paper describes an approach for training a humanoid controller that can mimic human motions. Simulated and real-world experiments are provided to demonstrate the controller's performance relative to a number of ablations. The controller is used to produce imitation learning data for several tasks and a policy is learned on top of the controller for each task.

**Summary Of Recommendation:**

Overall the paper has a good degree of novelty in the sense that we are at the leading edge of  sim2real learning for humanoids. The clarity of the paper can be improved as well as aspects of the evaluation. However, the novelty of the demonstrations makes up for these weaknesses.

---

### Official Review · Reviewer_2Dtf · 2024-07-21

**Originality:** 2
**Technical Quality:** 3
**Clarity Of Presentation:** 2
**Potential Impact:** 3
**Recommendation:** 3
**Confidence:** 5

**Review:**

Strengths:
- The universal interface supporting various input modalities (mocap, RGB, language) is a successful integration of many diverse components.
- The OmniH2O-6 dataset will provide robot data to the community for task/embodiment combinations that are not currently accessible.
- Compared to H2O, removing the requirement for mocap tracking during deployment allows the policy to be deployed outside of the lab.

Weaknesses:
- While improvements over H2O are claimed, the extent and source of these improvements are not clearly articulated.
- Robustness claims are not supported by comparisons or data.
- The role of history length and velocity estimation from history leave several open questions, see below.
- The demonstrated novelty from integrating various teleoperation interfaces with this controller is limited. The key capability of the controller is to imitate diverse whole-body motions, but the demonstrated behaviors might also be obtained using a decoupled controller for the legs/body.

**Quality Of The Limitations Section:**

2

**Questions For Rebuttal:**

A key claim of the paper is that OmniH2O "significantly improves upon prior art" H2O. Right now, it's hard to interpret which of the differences account for most of the performance. Is it possible to enumerate the exact difference between OmniH2O and H2O in a table with an ablation for each difference? As far as I can tell, the ablations (Table 1) already include the differences: [(1) number of keypoints, (2) lin vel observation], but there are also [(3) modified reward function, (4) modified domain randomization, (5) standing trajs added to the motion dataset, (maybe others?)]. Ablations 1 and 2 don't seem to explain much of the performance gap; can the improvement can be attributed to a specific design choice like a reward term?

Relatedly in Table 1, H2O and OmniH2O are evaluated on dataset $\hat{Q}$. Is this a fair comparison since OmniH2O was trained on $\hat{Q}$ but H2O was trained on a different dataset $Q$ and therefore being evalauted out-of-distribution?

The difference between the student with no history (OmniH2O-History0) vs. training with history (OmniH2O) is only 0.3%. This difference seems small; is it statistically significant across multiple seeds of training? If not, is it valid to assert that history utilization is an essential element? As it stands, I would be inclined to draw the opposite conclusion from this table and say that history length seems to have no trend / irrelevant for the OmniH2O policy. Therefore, the role of linear velocity observation would be mainly in easing the learning dynamics, and it does not need to be reconstructed from the history for humanoid locomotion. This is somewhat surprising since other works on bipedal (https://arxiv.org/pdf/2401.16889) and even quadruped locomotion (e.g. https://arxiv.org/abs/2301.10602, https://arxiv.org/abs/2202.05481) found a history module important for velocity estimation. Can you report the statistical significance in Table 1 and comment on this interpretation of the results?

Following on this point, Table 2 suggests that omniH2O outperforms omniH2O-History0 by a wide margin in the real world despite nearly identical simulation performance. Does this suggest that using a history of observations helps mitigate the sim2real gap? What would explain this since there does not seem to be an obvious mechanism for the history length to influence the sim2real quality while both are enjoying high performance on the training data?

“OmniH2O demonstrates great robustness under disturbances and unstructured terrains.” “OmniH2O shows superior robustness against human strikes and different outdoor terrains.” Superior to what? I am not sure how the reader is supposed to evaluate these claims. Is the robustness better than Unitree’s default controller, better than H2O, or other or cited works like https://arxiv.org/abs/2402.16796 or https://arxiv.org/pdf/2303.03381 ? The disturbance rejection looks somewhat similar to me. The locomotion in all videos takes place on flat surfaces and slight inclines which I do not believe should be referred to as unstructured terrains. What would you consider a structured terrain?

Does the teleoperation from verbal instruction w/ gpt-4 have significant differences from SayTap (https://arxiv.org/pdf/2306.07580) that prompts a VLM to output motion parameters for a quadruped and Code as Policies https://arxiv.org/abs/2209.07753 that prompts a VLM to output code?

Is there a fundamental advantage of whole-body control over a decoupled approach? It seems like all behaviors could potentially be teleoperated with an interface like https://arxiv.org/abs/2402.16796 which separates control of the arms and legs while also allowing the user to specify a base height command for squatting. It would strongly motivate the coupled training if OmniH2O is capable of some more dynamic movements involving more evident coordination between the upper and lower body. The "embodiment feasible" motion dataset appears to contain several motions like kicking and jumping that involve this type of coordination. Was sim-to-real a limitation for these motions? If so, maybe one could add videos of the robot doing tracking these moves in simulation?

**Robotics Focus:**

4

**Summary Of Paper:**

The paper presents a humanoid whole-body control RL policy, OmniH2O, and integrates it with various teleoperation methods. The RL training reward, observation, and architecture are modified to remove the body velocity observation while improving tracking performance compared to prior work. The release includes a teleoperation dataset recorded using the controller.

**Summary Of Recommendation:**

The paper is a nice demonstration of humanoid teleoperation and the claims/experiments open many interesting questions. The paper's clarity and novelty would be enhanced by more details on the difference between OmniH2O and H2O, evaluating the statistical significance of the history related claims, clarifying robustness claims, and showing whole-body motions that are not possible via decoupled control.

---

### Author Rebuttal · Authors · 2024-08-09

Dear Area Chair **N8rm** and Reviewers **2Dtf**, **ziut**, **huqu**,

Thank you for your thoughtful review of our manuscript, ID [303], titled *"OmniH2O: Universal and Dexterous Human-to-Humanoid Whole-Body Teleoperation and Learning"*. We truly appreciate the time and effort you put into providing constructive feedback.

We have carefully addressed each of the points you raised, and **we invite you to review our detailed responses in the rebuttal supplement PDF**. You can access it in the Openreview supplement or through the following anonymous link: [https://anonymous-omni-h2o.github.io/resources/rebuttal.pdf](https://anonymous-omni-h2o.github.io/resources/rebuttal.pdf)

Thank you once again for your valuable insights and for considering our work.

---

### Decision · Program_Chairs · 2024-09-04

**Decision:**

Accept

**Comment:**

The reviewers talk positively about the paper. The reviewers highlight that this is one of the first papers doing full-body mimicry for humanoids. The paper does many different qualitative experiments, including tennis, drawing, watering plants, and picking objects in indoor and outdoor environments. The main weaknesses are

* that the paper makes unsubstantiated claims about improving over previous work and the experiment complexity. The statements mainly exaggerate the contributions and shy away from limitations.

* the clarity of the paper can be improved. Especially the paper is partially overloaded the additional GPT4o and imitation learning experiments provide little value / insights. The paper would have been better without these experiments. The freed space should be used to elaborate on the limitations.

All in all, I agree with the reviewers and recommend accepting the paper. For the final submission, the authors should tone down the oversold statements and add a discussions about the limitations. There are still a lot of problems with the presented whole-body controller. The learned policies still have many undesirable properties such as tap dancing, the current controller kind of fails on dynamic tasks and all performed tasks do not require precise tracking. The paper does not need to solve all of these problems but it would be great to describe them and tell the reader where the performance would need to be improved and not list strawmen limitations in the limitation section.